# Brain Injury: Response to Injury Wound-Healing Mechanisms and Enlarged Perivascular Spaces in Obesity, Metabolic Syndrome, and Type 2 Diabetes Mellitus

**DOI:** 10.3390/medicina59071337

**Published:** 2023-07-20

**Authors:** Melvin R. Hayden

**Affiliations:** Diabetes and Cardiovascular Disease Center, Department of Internal Medicine, Endocrinology Diabetes and Metabolism, University of Missouri School of Medicine, One Hospital Drive, Columbia, MO 65211, USA; mrh29pete@gmail.com; Tel.: +1-573-346-3019

**Keywords:** astrogliosis, brain endothelial cells, enlarged perivascular spaces, microgliosis, perivascular macrophages, wound healing

## Abstract

Embryonic genetic mechanisms are present in the brain and ready to be placed into action upon cellular injury, termed the response to injury wound-healing (RTIWH) mechanism. When injured, regional brain endothelial cells initially undergo activation and dysfunction with initiation of hemostasis, inflammation (peripheral leukocytes, innate microglia, and perivascular macrophage cells), proliferation (astrogliosis), remodeling, repair, and resolution phases if the injurious stimuli are removed. In conditions wherein the injurious stimuli are chronic, as occurs in obesity, metabolic syndrome, and type 2 diabetes mellitus, this process does not undergo resolution and there is persistent RTIWH with remodeling. Indeed, the brain is unique, in that it utilizes its neuroglia: the microglia cell, along with peripheral inflammatory cells and its astroglia, instead of peripheral scar-forming fibrocytes/fibroblasts. The brain undergoes astrogliosis to form a gliosis scar instead of a fibrosis scar to protect the surrounding neuropil from regional parenchymal injury. One of the unique and evolving remodeling changes in the brain is the development of enlarged perivascular spaces (EPVSs), which is the focus of this brief review. EPVSs are important since they serve as a biomarker for cerebral small vessel disease and also represent an impairment of the effluxing glymphatic system that is important for the clearance of metabolic waste from the interstitial fluid to the cerebrospinal fluid, and disposal. Therefore, it is important to better understand how the RTIWH mechanism is involved in the development of EPVSs that are closely associated with and important to the development of premature and age-related cerebrovascular and neurodegenerative diseases with impaired cognition.

## 1. Introduction

The repair and remodeling of injured brain tissue during postnatal life are associated with the upregulation of some mechanisms that are characteristic of embryonic development [1]. These embryonic genetic memory mechanisms are present in brain cells and ready to be placed into action upon cellular injury. This genetically encoded mechanism may be termed the response to injury wound-healing (RTIWH) mechanism [2,3,4]. This recapitulated brain RTIWH mechanism consists of four basic phases: hemostasis, inflammation, proliferation, and remodeling, similar to peripheral wound-healing mechanisms [5,6]. However, the inflammatory and proliferative phases are unique in the brain, in that neuroglia play an active role [6]. The innate central nervous system (CNS) microglia and perivascular macrophages become reactive and undergo polarization in addition to the peripheral monocyte and monocyte-to-macrophage transformation. Because there are no fibrocytes/fibroblasts within the brain parenchyma, the astroglia become reactive and form an astroglia-derived gliosis scar instead of a fibrosis scar to protect the surrounding neuropil in injury states [6]. Additionally, brain injuries may be characterized as traumatic, radiation, metabolic, infectious, and hemodynamic stressors, and in this review the focus is primarily on metabolic injurious stimuli as occur in obesity, metabolic syndrome (MetS), and type 2 diabetes mellitus (T2DM) (Figure 1 and Figure 2) [1,2,3,4,5,6].

Perivascular spaces (PVSs) or Virchow–Robin spaces are fluid-filled spaces that ensheath the pial arteries, within the subarachnoid space (SAS) that dive deep into the white matter of the brain as ensheathed precapillary arterioles, where they end in true capillaries (Figure 3) [7].

The arterial and precapillary PVSs are bounded abluminally by the pia membrane that remains adherent to the pial arteries and arterioles, and is adherent to the adjacent basal lamina of the astrocyte endfeet basement membranes to form the glia limitans, as these structures penetrate the grey and deeper white matter regions of the parenchyma. Further, PVSs serve as the delivery conduit for the CSF to admix with the interstitial fluid (ISF) and aid to promote movement within the interstitial space (ISS). The true capillaries are without a pial lining and do not have a PVS. The true capillaries transition to the postcapillary venules, which are also ensheathed by a PVS (without a pia membrane), that form the efflux conduit for the ISF and metabolic waste, known as the glymphatic system [8] (abluminally lined by the basal lamina of the astrocyte endfeet), to exit the brain parenchymal ISS and travel to the SAS and, hence, to the CSF, to be delivered to the systemic circulation via the arachnoid granulations and the dural venous sinuses (Figure 3B,C) [7]. These PVSs are capable of enlarging or dilating and are termed as either enlarged perivascular spaces (EPVSs) or dilated PVSs, once they are 1–3 μm by T-2 weighted magnetic resonance imaging (MRI) [7,9]. EPVSs can be visualized and identified by non-invasive MRI, in addition to microscopic techniques including transmission electron microscopy (TEM), and are becoming more clinically important (Figure 4) [7,9].

EPVSs have been recognized as important remodeling changes in various neuropathologies [10]. EPVSs are known to associate with advancing age, hypertension, lacunes, microbleeds, intracerebral hemorrhages, cerebrocardiovascular disease with transient ischemic episodes and stroke, SVD, cerebral autosomal dominant arteriopathy with subcortical infarcts and leukoencephalopathy (CADISIL), cerebral amyloid angiopathy, obesity, MetS, T2DM, white matter hyperintensities, late-onset Alzheimer’s disease (LOAD), sporadic Parkinson’s disease, and non-age-related multiple sclerosis [7,9,10,11,12,13,14,15,16,17,18,19]. 

Cerebral small vessel disease (SVD) presents clinically as lacunar strokes, which are responsible for at least 20% of ischemic strokes, while representing a major cause of vascular cognitive impairment and vascular dementia. Importantly, EPVSs are known to be a biomarker and prominent feature of both SVD and VaD, which are known to be associated with lacunar stroke, in addition to white matter hyperintensities WMH [11,15,19,20,21]. EPVSs are also known to be a biomarker for SVD once identified by MRI and associate with white matter hyperintensities and lacunes [22]. Prior studies have shown that EPVSs associate with worse executive function and information processing in healthy older adults [23] and are significantly more prevalent in those with mild cognitive impairment as compared to age-matched control subjects without mild cognitive impairment [24]. These findings suggest that EPVSs may be an early remodeling change in the development of SVD and impaired cognition [23,24]. 

As our global aging population continues to grow, EPVSs are becoming increasingly important abnormal structural findings, since they also relate to clinical extracranial atherosclerosis [7,25,26], neurovascular cerebromacrovascular and cerebromicrovascular disease, and age-related neurodegenerative diseases such as LOAD and sporadic Parkinson’s disease. In older community-dwelling individuals free of clinical dementia and stroke, SVD biomarkers including EPVSs, white matter hyperintensities, and lacunes are related to worse cognitive performance [27]. Importantly, EPVS have recently been determined to be a marker for an increased risk of cognitive decline and dementia, independent of other small vessel disease markers over a period of four years [28]. 

The MetS is a cluster of multiple interconnected risks and variables, which associate with an increased risk for developing atherosclerotic cerebrocardiovascular disease and T2DM [25,29,30]. Importantly, visceral adipose tissue (VAT), insulin resistance, and MetS are each associated with an increased risk for developing EPVSs [31,32,33]. Insulin resistance is a central core feature of the MetS; however, it is a risk factor that may be independent of the MetS for developing EPVSs (Figure 5) [25,34].

There are four key features (hyperlipidemia, hyperinsulinemia, hypertension, and hyperglycemia) of the MetS, and each are interconnected to the central placement of insulin resistance (Figure 5) [25,29,30,34]. Globally, there exits an increase in obesity and the MetS due to an aging population, urbanization, sedentary lifestyles, and increased caloric high fat, sucrose, fructose, and glucose diets [25,29,30]. Further, our global population is now considered to be one of the oldest attained in our history [35]. Tucsek et al., have recently demonstrated that aging exacerbates obesity-induced neurovascular uncoupling, cerebromicrovascular capillary rarefaction, and cognitive decline in mice [36]. Thus, it is not surprising that aging is a reason for our observations of global increases in the MetS, EPVSs, and SVD [9,11,15,16,17,21,22,25,26,34,36].

Capillary rarefaction in the brain (loss of capillaries) has recently been found to be associated with an increase in obesity and the MetS [36,37,38]. Recently, Schulyatnikova and Hayden have hypothesized that capillary rarefaction may leave an empty space within the PVS that is subsequently filled with interstitial fluid [7]. This loss of capillaries within the PVS may allow for an increase in total fluid volume within the PVS when the capillary undergoes rarefaction and result in EPVSs (Figure 6) [7].

Capillary rarefaction (CR) is known to occur in multiple clinical situations, including: aging, hypertension, obesity, MetS, T2DM, and LOAD. Also, there are multiple proposed mechanisms that may co-occur to result in CR, including: oxidative–redox stress, inflammation, pericyte loss, BEC*act*/*dys*, impaired angiogenesis (increased ratio of increased antiangiogenic factors/proangiogenic factors), microvessel emboli and decreased microvessel shear stress, increased microvessel tortuosity, and, in some cases, increased TGF-beta [39]. 

While this mechanistic hypothesis for an expansion of the PVS is possible, more research will be required for it to gain support as a mechanism for increased EPVS. 

Obesity, MetS, and T2DM not only allow for the expansion of VAT depots to develop metainflammation and increased peripheral cytokines/chemokines (***p***CC) [7,25,34], but also provide a milieu for the development of gut microbiota dysbiosis with the secretion of ***p***CC, soluble lipopolysacchride (LPS), and LPS extracellular exosomes to result in the bidirectional communication between the gut and the brain ‘microbiota-gut-brain axis’ in neuroinflammation due to the excessive dual signaling of NVU BECs [7,25,34,40]. Importantly, there may be a host of metabolic functions, gut microbiota, and the innate immune system (Figure 7) [7,25,40,41,42,43].

This dual signaling by these two disparate regions allows for heightened signaling of the BEC to result in BEC*act*/*dys* as discussed in the following section [7,25,34].

## 2. Brain Endothelial Cell Activation and Dysfunction (BEC*act*/*dys*)

BECs and their luminal endothelial glycocalyx serve as the sentinel gatekeeper cells of the BBB and CNS as the first effector cell that comes into direct contact with peripheral systemic solutes (ions, molecules, neurotoxins) and cells, including leukocytes between the blood and the brain parenchyma (Figure 1) [44]. Importantly, BECs are key to vascular hemostasis, tone, leukocyte recruitment, hormone trafficking, and fluid movement from the blood to the parenchymal interstitial space, in addition to their playing a central role in the development of EPVSs [34,45]. Further, they are activated by the dual peripheral signaling from the ***p***CC and the peripheral soluble LPS and LPS-enriched small extracellular vesicles exosomes from the regional ***p***CC VAT depot metainflammation and the regional dysbiosis microbiota, respectfully (Figure 8) [7,34].

This dual signaling of BECs is associated with the increased expression of cell-surface adhesion molecules such a VCAM-1, ICAM-1, and E selectin, as defined in BEC*act*/*dys*. Likewise, BEC dysfunction is defined as the decreased synthesis, release, and/or activity of endothelium-derived nitric oxide (NO), which results in decreased bioavailable NO [7,34,44,46,47]. 

BEC*act*/*dys* is induced by: (1) proinflammatory cytokines, such as tumor necrosis alpha and interleukin-6; (2) turbulent blood flow, such as occurs at bifurcations and branch points of arteries; (3) advanced glycation end-products, which are elevated in hyperglycemia and aging; and (4) inflammatory stressors such as metainflammation ***p***CC, from VAT depots and plasma membrane peptides of gram-negative bacteria such as soluble LPS and LPS extracellular exosomes from gut microbiota dysbiotic regions. These four functions are each important mediators of EC*act*/*dys* via the activation of the nuclear transcription factor of BEC (Figure 8). The following TEM images are examples of BEC activation [25]. BEC*act*/*dys* is known to be induced by: (*i*) proinflammatory cytokines, such as tumor necrosis factor alpha and interleukin-6; (*ii*) turbulent blood flow, such as that which occurs at bifurcations and branch points of arteries; (*iii*) advanced glycation end-products, which are elevated in hyperglycemia and aging; and (*iv*) inflammatory stressors such as metainflammation and plasma membrane peptides of gram-negative bacteria such as soluble LPS and LPS extracellular vesicle exosomes. These four functions are each important mediators of BEC*act*/*dys* via the signaling of the BEC nuclear transcription factor. The following TEM images are examples of BEC*act*/*dys* (Figure 9) [7,25,34,46].

## 3. Hemostasis: Phase One

The homeostatic process of hemostasis is essential for any brain injury to its vasculature with hemorrhage, such as occurs with traumatic brain injury, microbleeds, and intracerebral hemorrhage. The initial response to this injury is constriction of the injured vessel with activation of platelets to form a fibrin clot and platelet plug. The fibrin clot results in hemostasis and also provides a scaffolding for the erythrocytes and incoming leukocytes of the inflammatory phase [48].

## 4. Inflammation—Neuroinflammation: Phase Two

Neuroinflammation is the 2nd phase of the RTIWH mechanisms, and if the CNS injurious stimuli are chronic and persistent as in obesity, MetS, and T2DM, the chronic sustained peripheral inflammation with ***p***CC, soluble LPS, and LPS extracellular vesicle exosomes injurious stimuli result in chronic activation of BECs with BEC*act*/*dys*. This chronic BEC*act*/*dys* results in chronic neuroinflammation with the generation of chronic CNS cytokines/chemokines [4]. BEC*act*/*dys* results in neuroinflammation, blood–brain barrier (BBB) disruption, and EPVSs in obesity, insulin resistance, and MetS (Figure 1 and Figure 7) [34]. Importantly, the systemic macrophages and the innate resident perivascular macrophages (rPVMΦs), the innate microglia cells (MGCs), are known to be the first set of effector cells in the RTIWH process, and considered to be the innate immune cells of the CNS [6,49,50].

The MetS, which includes obesity and T2DM, is widespread, globally affecting millions, and is known to cause the development of CNS neuroinflammation and neurodegeneration [25,51]. Indeed, the multiple components of the MetS are known to result in a wide range of effects, which include blood–brain barrier (BBB) disruption, neuroinflammation, EPVSs, cerebrocardiovascular disease, neurodegeneration, and impaired cognition (Figure 5) [25,34,51].

PVSs and EPVSs have become of increasing research interest, since EPVSs are associated with the early development of SVD and neurodegeneration that can be viewed by MRI noninvasively, in addition to the important role of PVSs, which form the conduit structure for clearance of waste via the glymphatic system [7,8,34]. The RTIWH remodeling is important for the formation of EPVSs via multiple mechanisms in the RTIWH process and entails the two-step process of neuroinflammation, as well as the myriad of other remodeling mechanisms (Figure 10) [7,34,51,52,53,54].

Brain PVMΦs belong to a distinct population of brain-resident myeloid cells located within the perivascular space that are derived from the yolk sac, similarly to microglia, and stain positive with CD-163 and lack the MGC specific marker P2RY12. PVMΦs also provide both structural and functional support for maintaining brain homeostasis, which includes maintenance of BBB integrity, glymphatic drainage, and mediation of immune functions including phagocytosis and antigen presentation [53]. The close ultrastructural (physical) relationship of the rPVMΦ within the postcapillary venule PVS allows for considerable cross-talk between the reactive perivascular macrophage (rPVMΦ), the NVU pericyte, and the astrocyte endfeet that line the abluminal PVS (Figure 11).

rPVMΦs mediate multiple mechanisms that contribute to remodeling changes of RTIWH to the PVS including: BBB disruption; increased permeability to fluids, solutes, and leukocytes; increased RONSS/RSI—oxidative/reductive stress; phagocytosis and increased phagocytic debris; *cns*CC—neuroinflammation; increased matrix metalloproteinases (MMP2 and 9); increased proteolysis; antigen presentation; EPVSs; and decreased waste clearance of the PVS conduit for the glymphatic system to the CBF [53,55].

According to Owens et al., the two-step process in the development of neuroinflammation depends heavily on the role of the postcapillary venular PVS [52]. The PVS provides the necessary space that leukocytes enter after undergoing diapedesis across the disrupted BBB that is instigated due to peripheral metainflammation and CNS neuroinflammation injury. Once the leukocytes have entered the PVS they then have to breech the outer abluminal glia limitans of the PVS, in order to arrive at their final destination, the ISS of the parenchymal neuropil and neuronal regions of the brain, via leukocyte MMP2 and MMP9 [52]. The initial inflammatory cells would be the neutrophiles, followed by the monocytes and lymphocytes. The innate immune leukocytes (neutrophiles and monocytes) and the acquired immune lymphocytes would then join the resident perivascular macrophages, and this combination of cells would then be capable of, not only causing the accumulation of cells and phagocytic cellular debris that causes PVS obstruction and the creation of EPVSs, but also create breaks in the abluminal glia limitans of the now-EPVS and be capable of crossing and entering the ISS of the parenchymal neurons [7,34,49,52,53,54].

There are basically two barriers to the passage of leukocytes into the parenchymal ISS of the CNS. The first barrier is the NVU BBB BECs of the postcapillary venule that is facilitated by cellular leukocyte adhesion molecules or vascular adhesion molecules, including intercellular adhesion molecules, vascular cell adhesion molecule-1, integrins, and selectins, which provide for the tethering, rolling, crawling, adhesion, and diapedesis via either the paracellular or transcellular routes, commonly referred to as transendothelial migration [56]. The second barrier is the glia limitans of the postcapillary venule PVS [52].

The author is becoming more confident that the inflammatory phase may be the most important or significant RTIWH phase in the development of the EPVS in obesity, MetS, and T2DM. Once BEC*act*/dys is initiated due to the constant injury by ***p***CC, soluble LPS, LPS extracellular vesicle exosomes, and subsequent *cns*CC, the inflammatory cells leave the postcapillary venule lumen and enter the PVS via a 2-step process, as previously described [52]. These incoming leukocytes join the rPVMΦs to become excessive and, along with the adaptive immune lymphocytes, fill up the PVS in addition to the excessive accumulation of their phagocytic debris that contribute greatly to the stalling and stagnation of flow to the point of obstruction, which results in EPVSs. This inflammatory component is in excess of the problems with impaired flow and pulsatility produced by vascular stiffening, which is another cause for impaired clearance of PVS conduit for glymphatic efflux of interstitial fluid and accumulating waste (including misfolded proteins such as amyloid beta and tau) and allows them to accumulate and undergo deposition and growth at sites of plaque niches within the interstitium [7,8,9,11,15,17,34,52].

Once there is stalling, stagnation, and obstruction of flow within the GS conduit, waste products accumulate along with proinflammatory leukocyte-derived cytokines/chemokines, and ROSs increase their neurotoxicity and the potential to cause even further injury to the tissue, with more RTIWH being placed into play to create a vicious cycle of continuous brain injury with RTIWH mechanisms [8,19,34]. In regards to LOAD, some view its early onset, development, progression, and acceleration to clinical symptomatic disease as being closely related to the vasculature, as has been described with mixed dementias with both neurovascular and neurodegenerative mechanisms occurring concurrently in the acceleration of impaired cognition and neurodegeneration [19]. Some even feel that the two primary problems in the development and progression of LOAD may be: number one, a disrupted and leaky BBB and number two, impaired clearance of accumulating waste, which may be now detected by non-invasive MRI studies. Thus, impaired clearance of waste via obstructed PVSs with ensuing EPVSs are an early remodeling change that is promoted by RTIWH [7,19,34].

## 5. Proliferation and Astrogliosis: Phase Three

Astrocyte(s) (ACs) have multiple homeostatic functions in the brain including their role as the connecting cell between the NVU and neurons creating the NVU coupling to increase cerebral blood flow when neurons increase their activity; their role as a major supplier of energy in the form of glucose and lactate via their capability for glycogen storage and glycolysis; a supplier of antioxidant reserves (glutathione and superoxide dismutase; and a supplier of various growth factors such as brain-derived growth factor and transforming growth factor-beta [57,58,59]. ACs also define many aspects of synapse formation, plasticity, and serve a protective function including synaptic maintenance and elimination or pruning along with microglia, since they are known to cradle the synapse [59,60]. It is important to note that human studies may not always conform and translate from rodent model studies, in that human ACs in the neocortical regions are much larger and these cytoplasmic processes extend much further, as compared to rodent models [61].

Brain injury results in reactive ACs (rACs) that are essential for early tissue protection. They are the second effector cell in the RTIWH process in the brain and are responsible for the proliferative phase and astrogliosis in the brain. Since the brain does not have fibrocytes/fibroblasts in the neuronal parenchyma, as occurs in non-neuronal tissues, there is no fibrosis within the parenchyma, but instead the rACs are responsible for forming gliosis and the glial scar [6]. Importantly, rACs instigate the cellular proliferative phase in brain injury, since neuronal cellular repair is impossible because CNS neurons are incapable of regeneration [6]. Thus, the proliferative phase occurs in the immediate areas surrounding regions of brain injury, and become reactive and undergo proliferation and hypertrophy, which is termed astrogliosis or gliosis, which is responsible for astrogliosis or glial scar formation to protect the normal surrounding neuronal tissues [6,7,34,61,62,63,64,65]. Further, the glial scar is known to replace the ischemic penumbra that originally surrounds the ischemic injury core (Figure 12) [63].

In addition to the formation of the brain injury glial scar, rACs are positioned as major regulators of vascular repair and remodeling following cerebral ischemia, stroke, and brain injury. For example, ablation of rACs disrupts vascular remodeling and worsens motor recovery following stroke by prolonging CBF deficits, increasing NVU permeability, and promoting apoptosis. Vascular remodeling, reorganization, and repair of the immediate surrounding region or penumbra of the ischemic death zone injury following stroke and the development of the glial scar are greatly dependent on rACs [66].

The transforming growth factor-beta family plays a critically important role in the formation of reactive ACs (rACs), gliosis, and the formation of the astroglia scar, and transforming growth factor-beta is primarily responsible for gliosis and glial scar formation in the brain [67,68]. In addition to being important in driving gliosis, transforming growth factor beta-1 also results in microvascular basement membrane remodeling with thickening due to increased AC fibronectin production following stroke, and this remodeling phase, in aged individuals. Further, increased transforming growth factor beta-1 results in decreased or stalling of CSF distribution within the PVS, impairs NVU coupling, and impairs neurologic recovery [68]. The predominant sources for brain transforming growth factor are the rMGCs and rPVMΦs following brain injury (stroke) and these cells, including the rACs, have all been shown to be capable of a marked upregulation of transforming growth factor beta-1 signaling following brain injury (stroke) [69].

### Astrocyte Aquaporin 4 (AQP4) Remodeling during Phase Three and Astrogliosis in Brain Injury

Among numerous roles of the ACs, they are important for controlling the volume of CNS, ISF, and PVSs, as well as the AC itself, via its highly polarized plasma membrane water channel AQP4, especially at the ACef in contact with the vasculature including the PVS [59,67]. The bidirectional water transport function of AQP4 is important in maintaining the homeostasis of water balance in the CNS. Upon forming rACs this AQP4 water channel is known to lose its ACef polarity at the plasma membrane and become maldistributed within the cytosol with loss of its normal function [59,66]. AQP4 is known to be widely expressed throughout the brain, especially at the blood–brain barrier, where AQP4 is highly polarized to astrocytic foot processes in contact with blood vessels. The bidirectional water transport function of AQP4 suggests its role in cerebral water balance in the CNS (Figure 13) [59,70].

## 6. Remodeling: Phase Four

To remodel means to change or alter a structure or form of tissue and RTIWH remodeling implies the rebuilding, repairing, and sculpting of tissues that are injured. Just as one remodels a damaged home, you must first tear down the damaged tissue via inflammation, proteolysis via matrix metalloproteinases, and phagocytosis to clean the injured region before you can begin to rebuild the injured tissue. Once the region is completely torn down and the damaged tissues are removed, the rebuilding or remodeling process can begin. Incidentally, the more tissue that is torn down, the more rebuilding or remodeling that will need to occur and the greater the remodeling. Interestingly, in addition to the professional phagocytes such as neutrophils and macrophages that assist in clearing the debris, rACs have been shown to be capable of phagocytosing debris including myelin [71,72].

Once the RTIWH phases have removed the damage-injured tissues, the remodeling phase begins and this entails the activation of multiple growth factors/hormones such as: the vascular endothelial growth factor family for vascular remodeling; brain-derived neurotrophic factor, which is related to the canonical nerve growth factor and is important in the support and survival of existing neurons, encouraging growth, and differentiation of new neurons and synapses; casein kinase 2, important for proliferation, growth cone formation, synaptic connections, and neuroprotective effects; growth-associated protein 43, which is also known as neuromodulin, involved in proliferation, synaptic plasticity, and growth; Kruppel-like factor-7, important in neuronal migration and differentiation; and transforming growth factor beta-1, which is important in growth, proliferation, and hypertrophy for neuronal limited regeneration remodeling [73]. Overall, the remodeling phase in the CNS is considered to be quite limited as compared to the non-CNS tissues since there is no long-term remodeling changes associated with fibrosis in the CNS due to lack of fibrocytes/fibroblasts [7].

Just beyond the glial scar that surrounds the ischemic necrotic core (Figure 12) there exists an encircling ischemic penumbra following ischemic stroke that is damaged but ‘not dead yet’ and may be salvaged with thrombolytic stroke therapy [74]. Notably, this ischemic penumbra is subject to remodeling and salvage via prompt recognition of clinical stroke and administration of either intravenous or intra-arterial tissue plasminogen activator therapy, whereby the multiple growth factors/hormones become activated and play a critically important role of neurovascular salvage in the remodeling phase, once an improved cerebral blood flow has been reestablished [73,75].

Upon completion of the remodeling phase in the RTIWH mechanism there occurs multiple remodeling changes, including EPVSs, histologically (previously presented and illustrated), and additionally, there will also remain EPVSs on T2-weighted MRI located in the basal ganglia and the centrum semiovale (Figure 14) [7,9,11,15,76].

Indeed, enlarged perivascular spaces (EPVSs) may exist in an evolutionary spectrum from postcapillary venule normal PVSs that are not enlarged to EPVSs in BG and CSO that associate with white matter hyperintensities and lacunes, to result in cerebral small vessel disease (SVD) vascular dementia with neuroinflammation, impaired cognition, and neurodegeneration in conditions of aging, hypertension, obesity, MetS, and T2DM [7].

Also, capillary rarefaction may be an end-point of the remodeling phase, which incorporates capillary loss especially, in obesity, MetS, and T2DM [7,32,37,38]. Importantly, during the process of capillary rarefaction an empty space will develop within the PVSs and allow for an increase in the total volume of the fluid-filled spaces within the PVSs and may contribute to dilated or EPVSs [7].

## 7. Resolution: Phase Five

Because the injurious stimuli are chronic in obesity, MetS, and T2DM, the RTIWH will continue and, therefore, it will not be possible for the brain to enter a complete resolution phase. However, if the injurious stimuli can be dampened or partially eliminated with proper treatment, the RTIWH resolution phase may ensue with limited resolution and even limited regeneration. This aspect of the RTIWH process may help to explain the improvement with diabetic end-organ remodeling with aggressive treatment to goal care plans.

## 8. Conclusions

In the introduction (Section 1), the first paragraph discusses brain injury and the RTIWH; the second paragraph discusses PVSs and the development of EPVSs as a result of the RTIWH; the third paragraph discusses the relation between obesity, MetS, T2DM, EPVSs, and the RTIWH process. Section 2 discusses the central importance of BEC*act*/*dys* in the development of EPVSs. Section 3 begins the discussion of RTIWH with hemostasis; Section 4 discusses the inflammatory phase of the RTIWH; Section 5 discusses the proliferation phase; Section 6 discusses the remodeling phase; Section 7 discusses the relative lack of a resolution phase in the RTIWH process in obesity, MetS, and T2DM. The examination of the processes involved in the development of EPVSs via the RTIWH mechanism is indeed a novel approach; however, it serves to enlighten us of this innate wound-healing mechanism in all individuals that is involved in brain injury and prepared to respond to injury at a moment’s notice. Importantly, it became obvious during this review that, during the development of EPVSs via RTIWH mechanisms, there exists a bidirectional relationship between PVSs and neuroinflammation, as described by Owens et al. [52], and there also exists a bidirectional relation between neuroinflammation and the development of EPVSs (Figure 10 and Figure 15) [77].

A central theme of brain injury involves gene activation and transcription of various factors in an attempt to respond to the multiple injurious stimuli, such that whenever there is injury the cells recapitulate their embryonic genetic memory in an attempt to heal through hemostasis, inflammation, proliferation growth (re-growth), hypertrophy, differentiation, development, remodeling, and limited neural regeneration. As a result of this chronically activated RTIWH mechanism, we as clinicians and researchers in this field of study must constantly review and expand our knowledge in an attempt to alter the RTIWH process in a manner that decreases the morbidity and mortality and the progressive brain-injuring nature of obesity, MetS, T2DM, and the associated accelerated atherosclerosis, along with their complications.

There has been increased interest in EPVSs recently, since the PVS is now recognized as the essential conduit for the glymphatic system for the removal of ISF and accumulated neurotoxic waste from ISS to the CSF, which acts as a sink for the accumulation of excess toxic waste from the CNS [8,78]. This recognition has brought forth an increased interest in the CNS interstitial space(s) (ISS) and how it empties into the PVS of the postcapillary venules. Even though the ISS diameter is only approximately 20–50 nanometers in mouse brains, its total volume is known to account for up to 20% of the total CNS volume and is crucial for interstitial fluid (ISF) solute transport, signal transmission, and communication amongst neurons (Figure 16) [79].

This neurotoxic metabolic waste may include the misfolded proteins of amyloid beta (40–42), tau, and alpha-synuclein, plus others that are known to accumulate in LOAD and sporadic Parkinson’s disease, respectively, within the interstitial fluid that is delivered to the subarachnoid space and eventually to the cerebrospinal fluid (CSF) for disposal. The finding that sleep is important in dilating the PVS to allow for improved drainage of ISF waste via the PVS–glymphatic system has recently been proposed [80,81,82]. Some have postulated that the vast majority of waste clearance occurs during sleep [81] and Zavecz et al. have recently viewed non-rapid eye movement sleep as a cognitive reserve factor in the presence of LOAD [82].

The importance of a healthy non-dilated PVS and its role as a conduit for the glymphatic system to deliver toxic metabolic waste from the interstitium and the ISSs to the CSF to provide a cleansing–flushing mechanism for the CNS and the remodeled dilated EPVS–Virchow–Robins space is not to be overlooked. The ability to view these abnormal EPVSs by non-invasive T-2 weighted MRIs provides an insight into the impairment of waste removal and its association with a RTIWH mechanism with impaired cognition, neuroinflammation, neurovascular and neurodegenerative remodeling diseases [28]. Indeed, EPVSs are not only a biomarker for SVD and associate with white matter hyperintensities, lacunes, aging, hypertension, obesity, MetS, and T2DM, but also according to the conclusion set forth by Paradise et al. [28], EPVSs may also be considered a marker for increased risk of cognitive decline and dementia, independent of other small vessel disease markers.

## Figures and Tables

**Figure 1 medicina-59-01337-f001:**
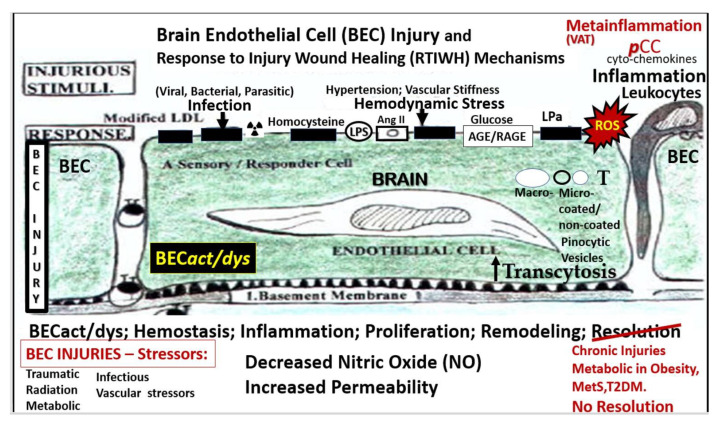
Multiple injurious stimuli to brain endothelial cells (BECs) in obesity, metabolic syndrome (MetS), and type 2 diabetes mellitus (T2DM). This representative illustration does not depict the thin luminal endothelial glycocalyx layer covering the BECs. Ang II = angiotensin 2; AGE/RAGE = advanced glycation end-products and its receptor RAGE; BEC = brain endothelial cell; BEC*act*/*dys* = brain endothelial cell activation and dysfunction; LDL = low-density lipoprotein cholesterol; LPa = lipoprotein little a; LPS = lipopolysaccharide; ***p***CC = peripheral cytokine/chemokine; ROS = reactive oxygen species and the reactive species interactome; T = transcytosis; 
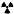
 = radiation symbol.

**Figure 2 medicina-59-01337-f002:**
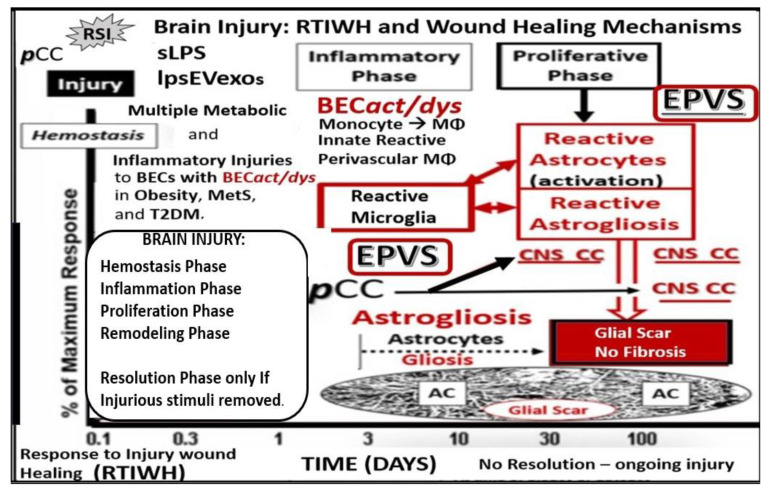
Comparison of peripheral response to injury wound healing (RTIWH) to the unique brain injury RTIWH mechanisms. This image depicts the RTIWH mechanisms as a result of the dual signaling by peripheral cytokines/chemokines (***p***CC), soluble lipopolysaccharide (sLPS), and LPS-enriched extracellular exosomes (lpsEVexos). This dual signaling of the brain endothelial cell(s) (BECs) results in BEC activation and dysfunction (BEC*act*/*dys*), which in turn results in central nervous system (CNS) neuroinflammation with increased *cns*CC, reactive polarized microglia, and reactive astroglia—reactive astrogliosis, which results in astrogliosis scarring instead of fibrosis scarring, since the brain does not have fibrocytes/fibroblasts in its parenchyma. Importantly, this RTIWH mechanism contributes to maladaptive remodeling and also contributes to the development of enlarged perivascular spaces (EPVSs). Figure adapted with permission by CC 4.0 [4].

**Figure 3 medicina-59-01337-f003:**
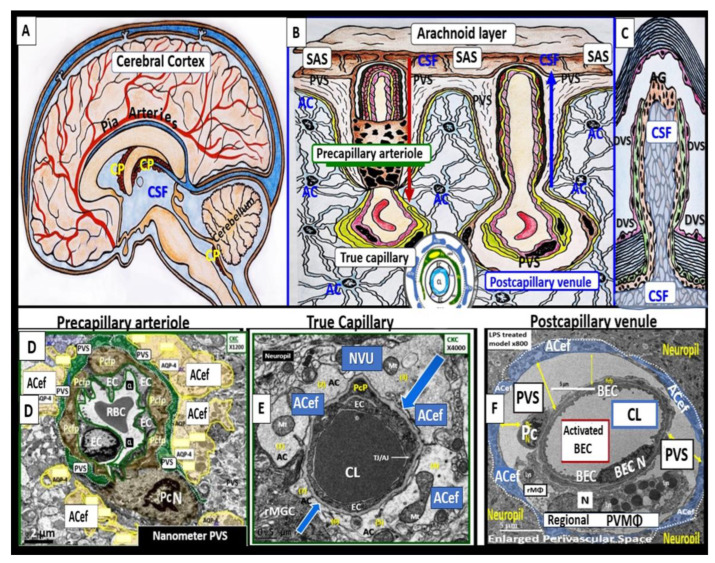
Illustration of the origins of the perivascular spaces (PVSs) and representative transmission electron micrograph (TEM) cross-sections of the precapillary arteriole, true capillary, and postcapillary venule. **Panel** (**A**) illustrates the whole brain with its surface pia arteries that dive deeply into the grey matter and deeply into the white matter, terminating in capillaries within the interstitial spaces (ISS). **Panel** (**B**) reveals the interaction of the PVSs to precapillary arterioles, which allow the vascular lumen influx of oxygen, nutrients, and solutes (red arrow), in addition to the influx of CSF via the PVSs (red arrow true capillaries with delivery of oxygen, nutrients, solutes, and the postcapillary venules that allow the removal of metabolic waste dioxide plus others), and PVSs that are responsible for the removal of the admixture of ISF and CSF, and metabolic waste (MW) that includes misfolded proteins (amyloid beta and tau neurofibrillary fibrils) (blue arrow) in the subarachnoid space (SAS). **Panel** (**C**) depicts how the CSF and its admixed metabolic waste from the PVS glymphatic system is transported to the dural venous sinus (DVS) and the systemic circulation. **Panels** (**D**–**F**) demonstrate representative and corresponding cross-sectional TEMs of corresponding labeled capillary images in **Panel** (**B**). **Panel** (**D**) demonstrates a precapillary arteriole with a PVS (pseudo-colored green). **Panel** (**E**) represents a true capillary without a PVS. Importantly, note that the pia membrane of the glia limitans abruptly ends at the true capillary and that the astrocyte endfeet end directly at the brain endothelial cell basement membrane (blue arrows) (numbers 1–7 indicate astrocyte numbers). **Panel** (**F**) depicts a postcapillary venule that appears to have a dilated PVS and is thus considered to be an enlarged PVS (EPVS). Note the presence of a resident perivascular macrophage cell within the EPVS. It is important to note that the PVS provides the conduit structure for the glymphatic draining system to deliver metabolic waste products to the CSF and the systemic circulation for disposal. This adapted and modified image is provided with permission by CC 4.0 [7]. *AC = astrocyte; ACef = astrocyte endfeet; ACfp = astrocyte foot process end-feet; AQP4 = aquaporin four; Cl = capillary lumen; CSF = cerebrospinal fluid; EC = brain endothelial cell; ISF = interstitial fluid; N = nucleus; Pc = pericyte; PVS = perivascular space; Mt = mitochondria; RBC = red blood cell; rMΦ = resident reactive macrophage*.

**Figure 4 medicina-59-01337-f004:**
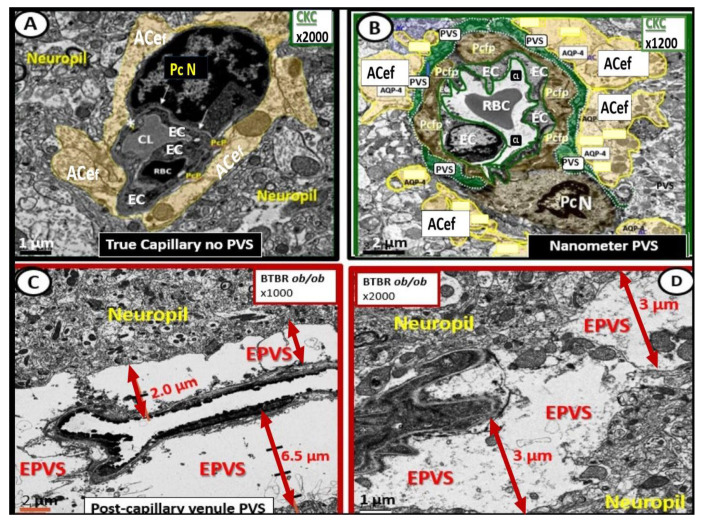
Comparison of normal true capillary and precapillary arterioles to postcapillary venules with enlarged perivascular spaces (EPVSs). **Panel** (**A**) demonstrates a true capillary with the astrocyte endfeet (ACef) abutting the shared basement membrane of the brain endothelial cell and pericyte (asterisk and closed arrows) without a pia mater membrane. **Panel** (**B**) demonstrates a precapillary arteriole with only a thinned perivascular space (PVS) (pseudo-colored green). Note that the abluminal perivascular space is still ensheathed by the pia matter and basal lamina of the adjacent ACef. **Panels** (**C**,**D**) each depict two different postcapillary venules with enlarged perivascular spaces (EPVS) between 2 and 6.5 μm. Modified image provided with permission by CC 4.0 [7]. Magnification ×1000; ×2000; scale bars = 2 and 1 μm in **Panels** (**C**,**D**) respectively. *AC astrocyte; ACef = astrocyte endfeet; AQP 4 = aquaporin 4; Cl = capillary lumen; EC = brain endothelial cell; Pc = pericyte; Pc N = pericyte nucleus; Pcfp = pericyte foot process—endfeet; RBC = red blood cell*.

**Figure 5 medicina-59-01337-f005:**
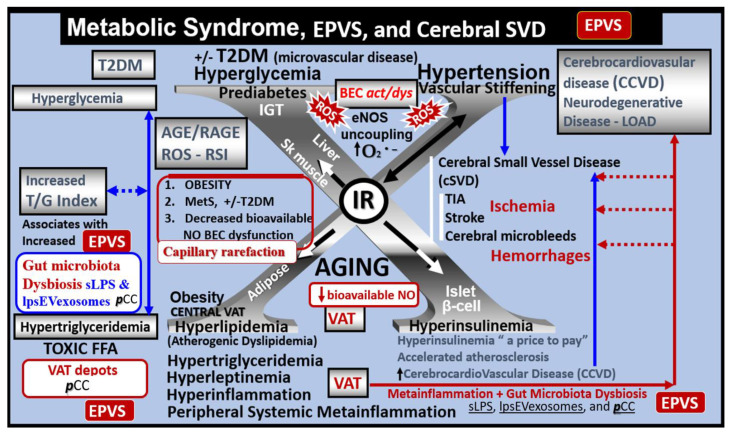
Metabolic syndrome (MetS), enlarged perivascular spaces (EPVSs), and cerebral small vessel disease (cSVD). The visceral adipose tissue (VAT), obesity, and hyperlipidemia (atherogenic dyslipidemia) located in the lower left-hand side of the letter X appears to drive the MetS and the other three arms of the letter X, which includes insulin resistance (IR) and the associated hyperinsulinemia, hypertension, vascular stiffening, and hyperglycemia, with or without manifest type 2 diabetes mellitus (T2DM). The global increase in obesity and the accumulation of visceral obesity is thought to be a major driver of the MetS that is related to the excessive metainflammation arising from VAT depots. Follow the prominent closed red arrows emanating from VAT to cerebrocardiovascular disease (CCVD), SVD, TIA, stroke, cerebral microbleeds, and hemorrhages. BEC activation and dysfunction (BEC*act*/*dys*), with its proinflammatory and prooxidative properties, result in endothelial nitric oxide synthesis (eNOS) uncoupling with increased superoxide (*O*_2_*^•−^*) and decreased nitric oxide (NO) bioavailability. Importantly, note that obesity, MetS, T2DM, and decreased bioavailable NO interact to result in capillary rarefaction that may allow EPVS development, which are biomarkers for cerebral SVD. Figure adapted with permission by CC 4.0 [7,25]. *AGE = advanced glycation end-products; RAGE = receptor for AGE; AGE/RAGE = advanced glycation end-products and its receptor interaction; βcell = pancreatic islet insulin-producing beta cell; FFA = free fatty acids—unsaturated long chain fatty acids; IGT = impaired glucose tolerance; LOAD = late-onset Alzheimer’s disease; ROS = reactive oxygen species; RSI = reactive species interactome; Sk = skeletal: TG Index = triglyceride/glucose index; TIA = transient ischemia attack*.

**Figure 6 medicina-59-01337-f006:**
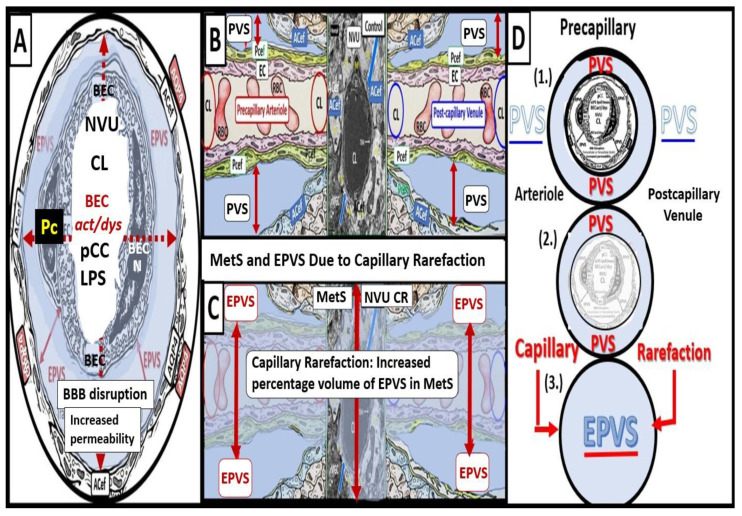
Cross- and longitudinal sections representative of pre- and postcapillary arterioles and venules with an encompassing surrounding perivascular space (PVS). **Panel** (**A**) depicts a cross-section of a capillary surrounded by a PVS (solid double red arrows) and its increase in total volume to become an enlarged perivascular space (EPVS) (dashed double red arrows), which represents capillary rarefaction. **Panel** (**B**) demonstrates a control longitudinal precapillary arteriole, postcapillary venule, and neurovascular unit (NVU) that runs through an encompassing PVS (light blue). **Panel** (**C**) depicts capillary rarefaction (CR) in a longitudinal view, and note how the volume of the PVS increases its total percentage volume once the capillary has undergone rarefaction as in obesity, metabolic syndrome, and type 2 diabetes mellitus. **Panel** (**D**) depicts the progression of a normal precapillary arteriole and postcapillary venule PVS to an EPVS once the capillary has undergone rarefaction, allowing for an increase in its total percentage volume of the PVS (1.–3.). **Panels** (**B**,**C**) provided with permission by CC 4.0 [7]. *ACef = astrocyte endfeet; AQP4 = aquaporin 4; BEC = brain endothelial cells; BECact/dys = brain endothelial cell activation and dysfunction; CL =capillary lumen; EC = endothelial cell; lpsEVexos = lipopolysaccharide extracellular vesicle exosomes; NVU = neurovascular unit; Pcef = pericyte endfeet*.

**Figure 7 medicina-59-01337-f007:**
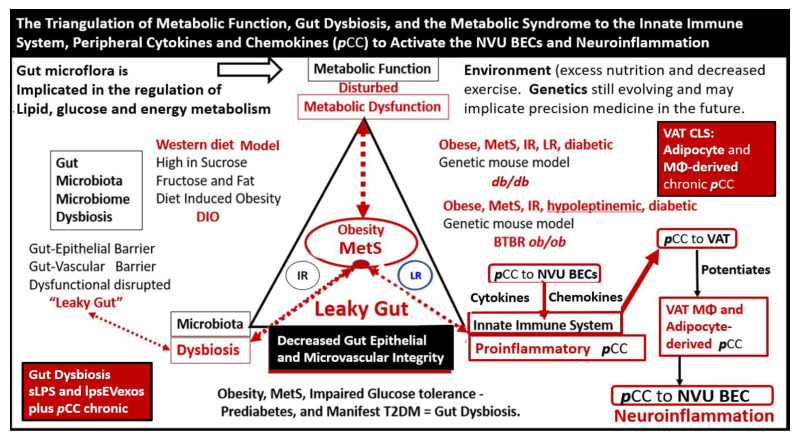
The triangulation of Metabolic function, gut dysbiosis, and the metabolic syndrome (MetS) allows the innate immune system, peripheral, proinflammatory cytokines and chemokines (***p***CC) to signal the neurovascular unit (NVU) brain endothelial cell(s) (BECs) resulting in BEC activation and dysfunction (BEC*act*/*dys*) with subsequent neuroinflammation. This schematic demonstrates that gut microbiota dysbiosis, metabolic dysfunction, and an activated proinflammatory innate immune system are associated with obesity and metabolic syndrome (lower-left red box). Importantly, this triangulation also associates with the metainflammation that is produced in the obese visceral adipose tissue depots (upper-right red box). These two distinct sites of metainflammation (red boxes) and dual signaling of BECs will each signal the central nervous system brain endothelial cells to result in neuroinflammation to result in BEC*act/dys* and contribute to enlarged perivascular spaces (EPVSs). Importantly, this dual signaling by the gut and visceral adipose tissue (VAT) of BECs may be synergistic in obesity and the MetS. Image provided with permission by CC 4.0 [34]. *Db/db = obese, insulin resistance diabetic genetic mouse model; DIO = diet-induced obesity; BTBRob/ob = black and tan brachyuric ob/ob mouse model of obesity and diabetes; IR = insulin resistance; LR = leptin resistance; sLPS = soluble lipopolysaccharide; lpsEVexos = lipopolysaccharide extracellular vesicle exosomes; MΦ = macrophage; MetS = metabolic syndrome; **p**CC = peripheral cytokines/chemokines; T2DM = type 2 diabetes mellitus*.

**Figure 8 medicina-59-01337-f008:**
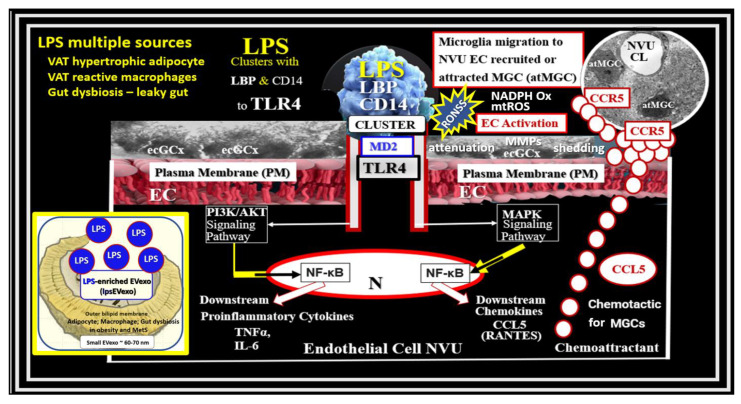
Dual Signaling of BECs by peripheral cytokines/chemokines (***p***CC) and soluble lipopolysaccharide (sLPS) and LPS-enriched small extracellular vesicle exosome (lpsEVexos), to result in brain endothelial activation and dysfunction (BEC*act*/*dys*). Note the insert lower-left, which depicts a small EVexo liberating multiple LPS-enriched vesicles and the metabolic signaling of BECs to result in central nervous system cytokines chemokines (*cns*CC) and attracted microglia cells (atMGCs) via CCL5 (RANTES). BEC*act*/*dys* results in neuroinflammation, blood–brain barrier (BBB) disruption, and enlarged perivascular spaces (EPVSs) in obesity, insulin resistance (IR), and the metabolic syndrome (MetS). This modified image is provided with permission by CC4.0 [34]. *atMGC = attracted microglia cell; CCL5 = chemokine (C-C motif) ligand 5 (chemoattractant for microglia cells); CD14 = cluster of differentiation 14; CCR5 = C-C chemokine receptor type 5, also known as CD195; CL = capillary lumen; EC = endothelial cell, brain endothelial cells; ecGCx = endothelial glycocalyx; EV = extracellular vesicles; EVMp =EVmicroparticles or microvesicles; EVexo = EVexosomes; IL1-β = interleukin-1 beta; IL-6 = interleukin-6; LPS = lipopolysaccharide; LBP = lipopolysaccharide-binding protein; MAPK = mitogen-activated protein kinase; Mp = microparticles; N = nucleus; PI3K/AKT = phosphatidylinositol 3-kinase/protein kinase B; PM = plasma membrane; RANTES = regulated on activation, normal T cell expressed and secreted; TNFα = tumor necrosis alpha*.

**Figure 9 medicina-59-01337-f009:**
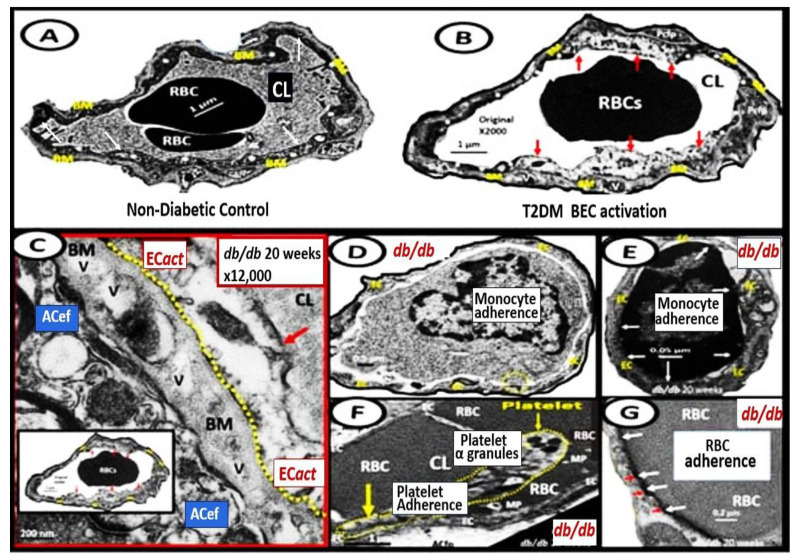
Representative transmission electron microscopic (TEM) images of endothelial cell activation (EC*act*). **Panel** (**A**) demonstrates a control BEC. **Panel** (**B**) depicts the thickened electron-lucent areas (red arrows) of EC*act* as compared to control models in **Panel** (**A**). **Panel** (**C**) depicts basement membrane (BM) thickening (red arrow) with increased vacuoles and vesicles (V). **Panels** (**D**,**E**) depict monocyte (**D**) and lymphocyte (**E**), platelet (**F**), and red blood cell (RBC) adhesion (**G**) to activated ECs (adhesions sites denoted by closed white arrows **Panels** (**E**,**G**); yellow arrows in **Panel** (**F**). Magnifications and scale bars vary. **Panels** (**C**–**G**) images are reproduced and modified with permission by CC 4.0 [25]. *ACfp = astrocyte foot processes*; *Cl, capillary lumen*; *EC = endothelial cells, brain endothelial cells*; *MP = microparticle of the platelet*.

**Figure 10 medicina-59-01337-f010:**
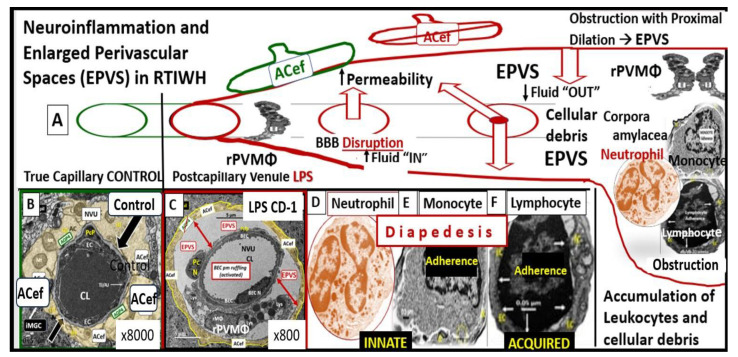
Response to injury wound-healing (RTIWH) remodeling, neuroinflammation, and enlarged perivascular spaces (EPVSs). **Panel** (**A**) illustrates the progression of the perivascular space (PVS) as it undergoes transformation—RTIWH remodeling to EPVSs in a linear fashion from left to right. **Panel** (**B**) demonstrates a transmission electron microscopic (TEM) image of a true capillary in a control 20-week-old female model. **Panel** (**C**) depicts a TEM image of an EPVS with a resident, reactive, migratory perivascular macrophage (rPVMΦ) in a 20-week-old female lipopolysaccharide-treated model. **Panels** (**D**–**F**) depicts leukocytes that undergo diapedesis across the neurovascular unit (NVU) brain endothelial cell(s) (BECs) disrupted blood–brain barrier (BBB), which enters the perivascular space(s) (PVS(s)). Along with the rPVMΦ, these leukocytes are capable of creating excessive phagocytic cellular debris. This increase in leukocytes and rPVMΦs along with their excessive cellular debris result in obstruction of the PVS and results in the dilation of the PVS proximal to the obstruction, creating the EPVS observed on magnetic resonance imaging (MRI) in the basal ganglia and centrum semiovale structures that appear early on in the development of cerebral small vessel disease and neurodegenerative diseases. While **Panels** (**A**–**F**) depict step one of a two-step process, step two incorporates the reactive oxygen species and the matrix metalloproteinases (MMP2, MMP9) that are capable of digesting the abluminal glia limitans of the EPVS, which allow for the transmigration of leukocytes to enter the interstitial spaces of the parenchymal neurons to complete the neuroinflammation remodeling mechanism in obesity, metabolic syndrome, and type 2 diabetes mellitus. Modified **Panels** (**A**,**B**) provided with permission by CC 4.0 [7,34] and **Panel** (**D**) by CC 4.0 [7].

**Figure 11 medicina-59-01337-f011:**
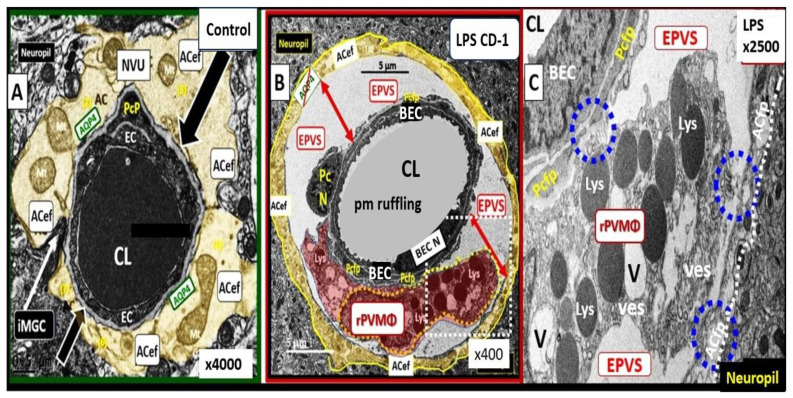
Enlarged Perivascular Spaces (EPVSs), reactive perivascular macrophages (rPVMΦs), and crosstalk with pericytes and foot processes (Pcfps) and PVS astrocyte endfeet (ACef). **Panel** (**A**) demonstrates a normal true capillary in a 20-week-old female C57B6/J control model and note how the ACef are arranged tightly about the shared basement membrane of the brain endothelial cell (BEC) and pericyte (black and white arrows). **Panel** (**B**) depicts an EPVS with an aberrant polarized rPVMΦ in a 20-week-old lipopolysaccharide (LPS)-treated male model and note how the ACef are markedly separated from the capillary (red double arrows). Also, note the white dashed-box bounds the area in **Panel** (**C**). **Panel** (**C**) depicts the communication crosstalk via rPVMΦs contacting the Pcfps and the EPVS abluminal lining ACfps (blue dashed-circles and white dashed-lines). Modified images provided by CC 4.0 [34]. *AQP4 = aquaporin 4; Lys = lysosomes; Mt = mitochondria; N = nucleus; NVU = neurovascular unit; V = vacuoles; ves = vesicles*. black/white arrows in subfigure (**A**) note how the ACef are arranged tightly about the shared basement membrane of the brain endothelial cell (BEC) and pericyte (black and white arrows). In Subfigure (**B**) white dashed boxed bounds the area in **Panel** (**C**) and the EPVS abluminal lining ACfps (blue dashed-circles and white dashed-lines.

**Figure 12 medicina-59-01337-f012:**
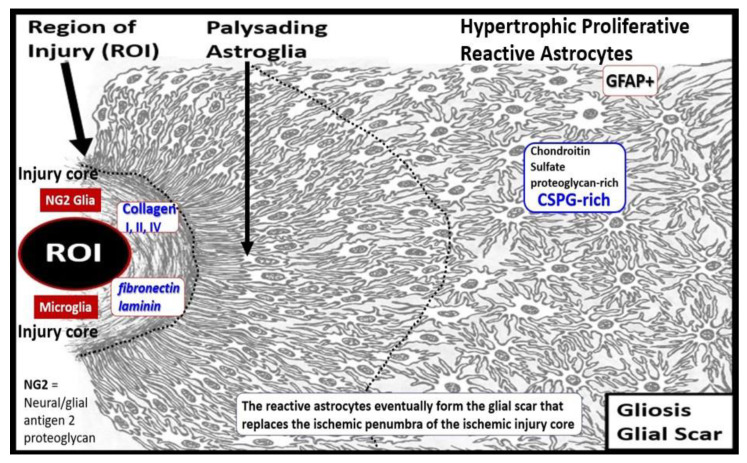
Response to injury wound healing (RTIWH) at regions of injury in the brain and formation of the glial scar. This illustration depicts the RTIWH at regions of brain injury that become surrounded with palysading astroglia that progress to hypertrophic, proliferative, and reactive astrocytes following brain injury, from left to right. Further, this gliosis scar serves to protect the surrounding uninjured brain structures from ongoing injury. This modified image is provided with permission by CC 4.0 [61].

**Figure 13 medicina-59-01337-f013:**
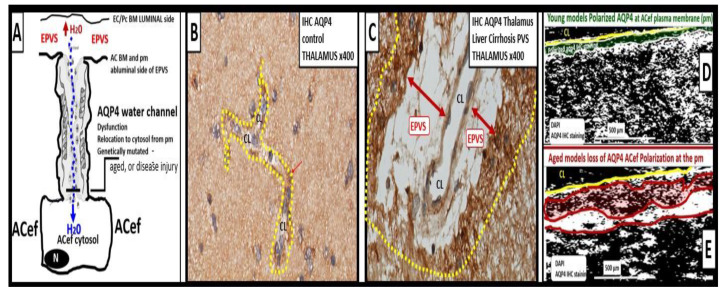
Representative illustrations of water channel aquaporin 4 (AQP4), and representative immunohistochemistry (IHC) staining of thalamus AQP4 in models. **Panel** (**A**) demonstrates an illustration of the AQP4 water channel localized to the astrocyte endfeet (ACef) at the plasma membrane that extends to the enlarged perivascular spaces (EPVS). **Panel** (**B**) illustrates an elongated capillary in the region of the thalamus in a control model and note the mild staining of the AQP4 via IHC staining with the AQP4 antibody (surrounded by dashed-yellow line). **Panel** (**C**) depicts marked heavily stained AQP4 water channels with IHC AQP4 antibody staining in a model with acetaminophen-induced liver failure and hepatic encephalopathy (surrounded by yellow-dashed line). AQP4 is known to be upregulated in aging and in ammonia-induced brain injury; however, it becomes maldistributed and there is a loss of plasma membrane (pm) polarity associated with its maldistribution and dysfunction. **Panel** (**D**) is a representative illustration of the AQP4 water channel that is polarized to the ACef pm (pseudo-colored green outlined by yellow line) that undergoes a loss of ACef polarization to become maldistributed in the cytosol of ACef in **Panel** (**E**). **Panel** (**E**) depicts the maldistribution of AQP4 IHC staining (pseudo-colored red outlined by yellow line) and note the increased condensation as compared to the more flocculent IHC staining **Panel** (**D**). Representative illustrations in **Panels** (**A**,**D**,**E**) were created in Microsoft 3-D paint program. Modified in **Panel** (**A**) image reproduced with permission by CC 4.0 [34]. Note the insertion at end of **Panels** (**B**,**C**) (surrounded by dashed-yellow line). Also note in **Panel** (**D**) (pseudo-colored green outlined by yellow line) **Panel** (**E**) (pseudo-colored red outlined by yellow line) *BM = basement membrane; CL = capillary lumen; DAPI = 4′,6-diamidino-2-phenylindole fluorescent stain; EC/Pc = endothelial cell/pericyte; H_2_O = water; pm = plasma membrane*.

**Figure 14 medicina-59-01337-f014:**
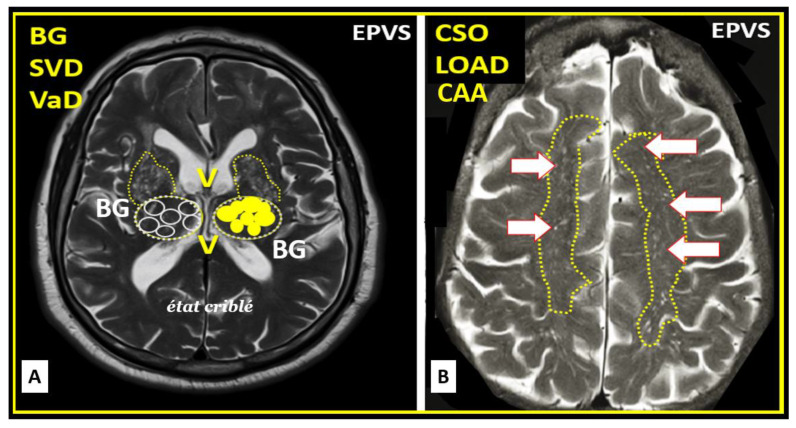
Magnetic resonance imaging (MRI) comparisons of basal ganglia (BG) enlarged perivascular spaces (EPVSs) to centrum semiovale (CSO) EPVSs. **Panel** (**A**) depicts the paired EPVSs within the BG that are traced in open on the left and masked yellow on the right BG. Note the white spaces within the paired dashed lines just above the paired BG structures. MRI image from a 75 y/o male status post-stroke, recovered with small vessel disease. **Panel** (**B**) depicts the paired elongated oval structures outlined by yellow dashed lines to enclose multiple white enlarged perivascular spaces. Note the open white arrows outlined in red pointing to prominent EPVSs. MRI image from a 79 y/o female with history of transient ischemic attacks. Importantly, note that BG EPVSs are strongly associated with cerebral small vessel disease (SVD) in **Panel** (**A**) and that CSO EPVSs are strongly associated with late-onset Alzheimer’s disease and cerebral amyloid angiopathy (CAA) in **Panel** (**B**).

**Figure 15 medicina-59-01337-f015:**
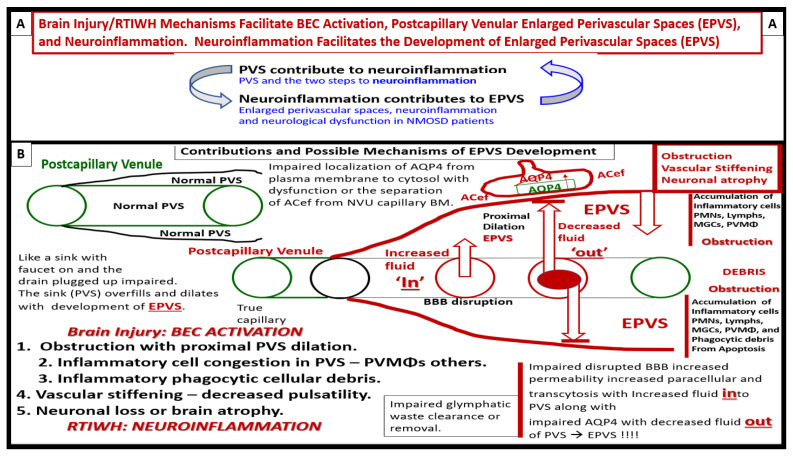
Brain injury and the response to injury wound healing (RTIWH) facilitates brain endothelial cell (BEC) activation, neuroinflammation, and enlarged perivascular spaces (EPVSs). **Panel** (**A**) illustrates the bidirectional relationship between perivascular spaces (PVSs) and neuroinflammation in relation to EPVSs. **Panel** (**B**) demonstrates the contribution and mechanisms to the development of EPVSs. The brain is exposed to many injurious stimuli over time and in multiple diseases and syndromes such as obesity, metabolic syndrome, and type 2 diabetes mellitus. These injurious stimuli result in BEC activation and dysfunction (BEC*act*/*dys*), which acts as a pivotal point in the RTIWH and these mechanisms also contribute to multiple wound-healing phases (hemostasis, inflammation–neuroinflammation, proliferation, remodeling, and resolution, only if the injurious stimuli are damped down or removed). Since these injurious stimuli are chronic, the brain remains under the constant influence of the wound-healing phases and chronic remodeling of essential brain structures and the development of EPVSs. Herein we propose a sequence of events in the development of EPVSs as follows: Brain injury, BEC*act*/*dys*, neuroinflammation with RTIWH, obstruction of PVSs due to accumulation of inflammatory cell leukocytes within and around the neurovascular unit and postcapillary venular PVS and inflammatory-derived cellular debris, vascular stiffening with deceased pulsatility, and neuronal degeneration with neuronal loss and brain atrophy.

**Figure 16 medicina-59-01337-f016:**
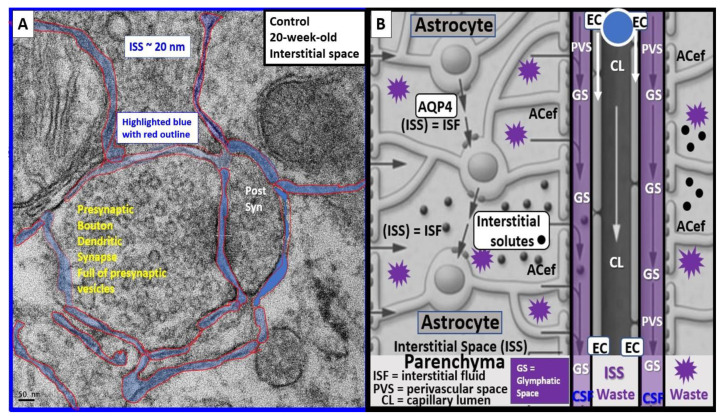
Transmission electron microscopic image of interstitial spaces (ISS) and illustration of interstitial fluid flow to the glymphatic system. Interstitial spaces (ISS) are nanometer structures; however, they combine to create a total of 20 percent of total brain volume and empty their interstitial fluid contents and metabolic wastes into the perivascular conduit spaces of the glymphatic space, or system (GS) to be delivered to the cerebrospinal fluid (CSF). **Panel** (**A**) demonstrates the nanosized diameter of ISS containing the ISS in a control 20-week-old mouse model from the frontal cortical grey matter. **Panel** (**B**) depicts how the ISS containing the ISF, solutes (black dots), and metabolic waste empties into the post-capillary venule perivascular space (PVS) or the conduit of the GS, which is responsible for toxic waste clearance. *AC = astrocyte; ACef = astrocyte endfeet; EC = endothelial cell; CL = capillary lumen*.

## Data Availability

Data and materials will be provided upon reasonable request.

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
