# Peer review of "Brain Injury: Response to Injury Wound-Healing Mechanisms and Enlarged Perivascular Spaces in Obesity, Metabolic Syndrome, and Type 2 Diabetes Mellitus"

_medicina, 2023, doi:10.3390/medicina59071337_

Round 1
Reviewer 1 Report
The topic of the manuscript is interesting, however the approach is very general and difficult to follow.
Images must be improved in technical quality but also design because are crowded and are do not offer a selfexplanation. Could be better if the author use diagrams and not micorscopy images. Also the figure foots are very long and there is not a correspondance with the main text.
There is a lack of discussion, so the manuscript looks like a large compilation of information without message.
I suggest to focuse the review in one topic: anatomical and celular changes with different metabolic insults; common celular mechanism ocurring with metabolic insults and responsibles of morphological changes; or an integrated mechanism of RTIWH.
Minor revision
Author Response
Response to Reviewer number 1 round 1
Regarding: Manuscript ID: medicina-2477773
Dear reviewer number 1,
First, I wish to thank you for taking your time, utilizing your effort, and sharing your knowledge in order to review this complicated submitted manuscript. My responses to your suggestions and recommendation will be in blue color in this response and also in the text.
Comments and Suggestions for Authors
The topic of the manuscript is interesting,
Author wishes to thank reviewer number 1 for this kind comment.
however the approach is very general and difficult to follow. While the manuscript may be difficult to follow the topic and approach in regards to brain injury and the response to injury wound healing mechanisms may help us to better understand this emerging science regarding the development of enlarged perivascular spaces (EPVS).
Images must be improved in technical quality but also design because are crowded and are do not offer a selfexplanation. Author has rebuilt multiple figure images and improved both the clarity of the images as well and improving the labeling so readers should now be able to better understand these images and their labels. Please note the response to reviewer number 2 for the exact figures involved, and the revised submitted manuscript.
Could be better if the author use diagrams and not micorscopy images. Also the figure foots are very long and there is not a correspondance with the main text. Author has shortened some of the figure legends and abbreviations. The figure legends now appear to have a better correspondence with the text. See revised manuscript.
There is a lack of discussion, so the manuscript looks like a large compilation of information without message.
Author has attempted to discuss each of various steps in discussing each of processes involved in the response to injury mechanisms and how they are related to the development of enlarged perivascular spaces (EPVS), i.e. beginning with 1. Introduction: wherein author provides the background to the problem and help the readers to better understand the importance of perivascular spaces and the development of enlarged perivascular spaces utilizing seven figures and illustration; followed by section 2. Brain endothelial cell activation and dysfunction (BECact/dys); followed by section 3. Hemostasis: Phase One, the first phase of the brain response to injury wound healing; followed by section 4. Neuroinflammation: Phase two; followed by Section 5. Proliferation and Astrogliosis: Phase Three; followed by Section 6. Remodeling: Phase Four; followed by Section 7. Resolution: Phase Five; followed by Section 8: Conclusion. Additionally, since this was a review the closing remarks were presented in the conclusion since this manuscript was not an article that presented research data, but a narrative review.
I suggest to focuse the review in one topic: anatomical and celular changes with different metabolic insults; common celular mechanism ocurring with metabolic insults and responsibles of morphological changes; or an integrated mechanism of RTIWH.
Author agrees that these suggestions are very good and thoughtful but I wish to stay with the outline and review as in the submitted manuscript since I wished to share with readers that the concept of brain injury and response to injury wound healing mechanisms play a key role in the development of enlarged perivascular spaces (EPVS). Maybe in future papers I can make good use of reviewer number 1 excellent suggestions.
Sincerely, with gratitude and appreciation,
Melvin R Hayden
Submitting author

Reviewer 2 Report
The manuscript is well written. All possible pathways happened in response to brain injury and wound healing and their pathological consequences have been discussed comprehensively. However, the following flaws need to be corrected.
1- The Figures are blurred. labeling in small font is completely unreadable in most of the figures. For example; fig 3, panels D, E, and F, fig 4 panel B, Fig 6 panels B and C, fig 8, fig 9. Color contrast and font size of labelings need improvement.
2- Figures 10, 11, 13, 14, and 16 have noot been cited in their labelled description.
3- Fig 15; labelling description ends at two ful stops (. .).
4- Page 16, 2nd and 3rd paragraphs should be cited.
5- Page 17, third last row, citation numbers (6, 7. 34, 61, 62, 63, 64, 65); I guess there must be (,) in place of (.) between 7 and 34.
Author Response
Response to Reviewer number 2 round 1
Regarding: Manuscript ID: medicina-2477773
Dear reviewer number 2,
First, I wish to thank you for taking your time, utilizing your effort, and sharing your knowledge in order to review this complicated submitted manuscript. My responses to your suggestions and recommendation will be in blue color in this response and also in the text.
The manuscript is well written. All possible pathways happened in response to brain injury and wound healing and their pathological consequences have been discussed comprehensively. However, the following flaws need to be corrected.
Author wishes to thank the reviewer for these kind comments.
1The Figures are blurred. labeling in small font is completely unreadable in most of the figures. For example; fig 3, panels D, E, and F, fig 4 panel B, Fig 6 panels B and C, fig 8, fig 9. Color contrast and font size of labelings need improvement.
Author agrees with reviewer number 2 and has carefully rebuilt the above figures to improve their clarity and relabeled the images to improve them. Please see revised manuscript. Author did notice that when I converted the word document to PDF that some clarity to figures did dimmish. Consider looking at the word document to better appreciate the revised figures. Or examine the PDF views of the revised figures.
2- Figures 10, 11, 13, 14, and 16 have noot been cited in their labelled description.
These figures have now been labeled and are now cited as to their origins in each figure legend.
3- Fig 15; labelling description ends at two ful stops (. .). This was corrected and please see revised manuscript
4- Page 16, 2nd and 3rd paragraphs should be cited. Author has now added additional appropriate references/citations to these 2 paragraphs
5- Page 17, third last row, citation numbers (6, 7. 34, 61, 62, 63, 64, 65); I guess there must be (,) in place of (.) between 7 and 34
Author has now corrected this flaw, please see revised manuscript.
Sincerely with gratitude and appreciation
Melvin R Hayden
Submitting author

Round 2
Reviewer 1 Report
The quality of the figures and style of the manuscript "Brain Injury: Response to Wound Healing Mechanisms from Injury and Enlarged Perivascular Spaces in Obesity, Metabolic Syndrome, and Type 2 Diabetes Mellitus" has been improved. However, it seems to be focused on a general anatomical description of the phenomenon and the information is mainly refered to the process during senescence and neurodegenerative diseases. Figure captions contain more information than the text itself. The graphic summary, figures 1, 2, 9, 10 and 11 are still difficult to read due to quality. I recommend restructuring the information.
Author Response
Response to Reviewer #1 Round 2
The quality of the figures and style of the manuscript "Brain Injury: Response to Wound Healing Mechanisms from Injury and Enlarged Perivascular Spaces in Obesity, Metabolic Syndrome, and Type 2 Diabetes Mellitus" has been improved.
Thank you very much.
However, it seems to be focused on a general anatomical description of the phenomenon and the information is mainly refered to the process during senescence and neurodegenerative diseases.
I chose to concentrate on ultrastructural – fine structure anatomy because my primary skill is in the field of transmission electron microscopy and I chose the focus on senescence and neurodegenerative disease because we are talking about individuals with obesity, metabolic syndrome and T2DM who are at an increased risk for the development enlarged perivascular spaces because they have so much macro and microvascular disease . Plus, we are now living in one of the oldest global societies recorded according to the WHO.
Figure captions contain more information than the text itself.
I tried to go back and trim some of these down however, as I was doing this there seemed to be too importance being lost.
The graphic summary, figures 1, 2, 9, 10 and 11 are still difficult to read due to quality. I recommend restructuring the information.
I have rebuilt figures 1, 2, 9, 10, and 11. Thank you again for taking your time to review this submission and I really appreciated your inputs and suggestions. It was my privilege and honor to work with you.
Submitting author
Melvin R Hayden
University of Missouri School of Medicine
Columbia, Missouri USA
d 2
